# Experiences of HPV self-collection among Aboriginal and Torres Strait Islander women and people with a cervix

**Louise E. Mitchell**[1]*, **Emily Phillips**[1], **Chloe J. Jennett**[2], **Claire Bavor**[3],
**Tessa Saunders**[3], **Claire Nightingale**[3], **Megan A. Smith**[2], **Lisa J. Whop**[1], on behalf of
**Screen Your Way Investigator Team**¶

1 Yardhura Walani, National Centre for Aboriginal and Torres Strait Islander Wellbeing Research, Australian National University, Canberra, Australian Capital Territory, Australia, 2 Sydney School of Public Health, The University of Sydney, Sydney, New South Wales, Australia, 3 Evaluation and Implementation Science Unit, Centre for Health Policy, Melbourne School of Population and Global Health, University of Melbourne, Melbourne, Victoria, Australia

¶ Membership of the Screen Your Way Investigator Team is provided in the Acknowledgements.
* louise.mitchell@anu.edu.au

## Abstract

Since July 2022, Australian guidelines have recommended that anyone eligible for cervical screening be offered a choice between using a self-collected vaginal sample or a clinician-collected cervical sample for Human Papillomavirus (HPV) testing. This study explored cervical screening among 555 Aboriginal and Torres Strait Islander women and people with a cervix, 261 who had screened since the policy change ('recently screened'). Participants were recruited for an online survey between December-2023 and April-2024. Over half of recently screened participants were offered a choice of collection methods (n = 151, 58%). Of those offered the choice, 67% chose to screen using self-collection (n = 101). In total, 46% (n = 118) of recently screened participants used self-collection, either themselves at home (n = 41, 35%) or the clinic (n = 48, 41%) or assisted by a healthcare provider without a speculum (n = 29, 24%). Among those who collected their own sample (n = 89), the main reasons were it was less embarrassing, they felt in control of their body, and it was less scary. However, only 55% of these participants felt they had enough information to make an informed decision between collection methods. Over half of recently screened participants reported having a clinician-collected sample with a speculum (n = 133, 51%). The majority were not offered a choice of collection method (61%, n = 81), however 38% (n = 50) were and chose a clinician-collected sample. The main reasons for choosing a clinician-collected sample included always having had it done by a healthcare provider, wanting the healthcare provider to have a look or believing the healthcare provider would collect a better sample. This study highlights a preference for HPV self-collection among Aboriginal and Torres Strait Islander women and people with a cervix, including those who already participate in cervical screening.

**Data availability statement:** In accordance with principles of Indigenous Data Sovereignty, the research team remains the custodian of the data, and Aboriginal and Torres Strait Islander participants retain ownership. The data collected in this study includes individual survey responses which contain potentially identifying and sensitive information. Participants did not consent to public release of their data, and access is subject to ethical restrictions imposed by the Human Research Ethics Committees (HRECs) listed in the Ethics Approval section. Requests for access to the minimal de identified dataset reviewed in consultation with Thiitu Tharrmay and may be directed to Yardhura Walani, Australian National University (email: yardhurawalani@anu.edu.au; phone: +61 2 6125 5111), or the Australian Institute of Aboriginal and Torres Strait Islander Studies (AIATSIS; email: ethics@aiatsis.gov.au; phone: +61 2 6246 1111).

**Funding:** This research has been funded through two Australian National Health and Medical Research Council (NHMRC) Targeted Call for Research competitive funding grants (LJW: GNT201490, CN: GNT201578). LJW is supported by a NHMRC Investigator Grant (2009380) CN is supported by a Mid-Career Research Fellowship (MCRF21039) from the Victorian Government acting through the Victorian Cancer Agency. CB is supported by a NHMRC Postgraduate Scholarship (GNT2030732) and an Australian Government Training Program Scholarship. CJJ is supported by a scholarship from the University of Sydney. The funding organisations have no role in the collection, management, analysis, and interpretation of data; writing of the protocol or subsequent manuscripts; and the decision to submit the protocol for publication. The views expressed in this publication are those of the authors and do not necessarily reflect the views of the funders.

**Competing interests:** The authors have declared that no competing interests exist.

**Abbreviations:** CST, cervical screening test; HCP, healthcare provider; HPV, Human Papillomavirus; NCSP, National Cervical Screening Program.

It reinforces the importance of offering all eligible participants a choice of collection methods and supporting informed decision-making.

## Introduction

Cervical cancer is preventable and can be effectively treated if detected early [1,2]. Australia offers both primary and secondary prevention programs: a school-based human papillomavirus (HPV) immunisation program and primary HPV screening through the National Cervical Screening Program (NCSP). As a consequence of these programs, Australia will potentially become the first country to actively eliminate cervical cancer as a public health problem [2,3]. However, inequities remain and without targeted efforts, elimination may not include Aboriginal and Torres Strait Islander women and people with a cervix [4]. Cervical cancer burden among Aboriginal and Torres Strait Islander women and people with a cervix remains unacceptably high, with age-adjusted incidence rates 2.1 times higher (2016–2020) and mortality rates 2.8 times higher (2018–2022) compared to non-Indigenous individuals [5].

The HPV vaccine has been available since 2007, initially targeting girls aged 12–13 and expanding to include boys in 2013. The program transitioned from a three-dose quadrivalent vaccine to a two-dose nonavalent schedule in 2018, and in 2023, adopted a single-dose based on evidence of one-dose efficacy [6,7]. While HPV vaccination rates among Aboriginal and Torres Strait Islander adolescents peaked in 2020, they have since declined, with 2023 data showing 80.9% of girls and 75.0% of boys were vaccinated before age 15 [8]. The shift to a one-dose schedule aims to support more equitable vaccine coverage.

In December 2017, the NCSP transitioned from cytology-based (Pap test) to HPV DNA-based screening, inviting women and people with a cervix aged 25–74 years to undertake a cervical screening test every five years. In 2023, among Aboriginal and Torres Strait Islander people aged 25–74, HPV 16/18 was detected in 2.2% of tests and other oncogenic HPV types in 9.3%. Those eligible for HPV vaccination (born after 30 June 1980) had lower positivity for HPV 16/18 (1.6%) but higher rates for other oncogenic types (13.7%), compared to those not offered vaccination (born on or before 30 June 1980), who had 2.7% and 5.2% positivity respectively. These patterns reflect the vaccine's targeted protection against HPV 16/18 and the higher prevalence of HPV in younger, vaccinated age groups [5].

The transition to primary HPV testing also allowed for the introduction of HPV self-collection, which allows women and people with a cervix to collect a vaginal screening sample using a flocked swab to be tested for HPV. Initially, self-collection was thought to be less sensitive than clinician-collection, so it was limited to individuals aged over 30 years who were two or more years overdue or never-screened, and declined clinician testing [9]. These restrictions led to low uptake and poor awareness among both clinicians and those eligible for screening [10]. The number of Aboriginal and Torres Strait Islander people to have used self-collection under this model is unknown.

In July 2022, based on strong evidence of the equivalent sensitivity between self-collection and clinician-collection, and modelling suggesting benefits from improved participation would outweigh any feasible loss in sensitivity [11], Australia updated its policy to expand eligibility for self-collection to include anyone eligible for cervical screening within the context of a practitioner-supported model of care (referred to as 'universal access'). [12,13]. This change aimed to improve access in practice to self-collection, and thereby remove barriers to cervical screening participation and reduce inequities in cervical cancer incidence and mortality [14]. However, it is unknown to what extent this choice is being offered as a result of the policy change. Research has demonstrated initially slow implementation with variable knowledge and adoption among the primary care workforce resulting in inconsistent promotion of self-collection [15].

Universal access to self-collection provides an opportunity to address screening inequities among Aboriginal and Torres Strait Islander women and people with a cervix. However, it must not be implemented in isolation. Without addressing systemic issues, such as racism, access to culturally safe care, and the ongoing impacts of colonisation, inequities in cervical screening participation will persist. While NCSP data is not yet available for Aboriginal and Torres Strait Islander women and people with a cervix who have used self-collection, research suggests it is highly acceptable. By offering increased privacy, comfort, convenience and autonomy, self-collection has the potential to increase cervical screening participation [16–18]. It remains uncertain whether Aboriginal and Torres Strait Islander screening participants are adequately informed about self-collection as a choice for cervical screening, or what their experience has been in accessing self-collection since the implementation of universal access.

This study aimed to understand the screening practices of Aboriginal and Torres Strait Islander women and people with a cervix and explore the experiences of recently screened participants since the policy change to introduce universal access to HPV self-collection (i.e., since July 2022).

## Materials and methods

### Positionality

This work is led by a research team with Aboriginal and Torres Strait Islander lived experience (LJW, EP), Aboriginal and Torres Strait Islander leadership in public health (LJW, EP), epidemiology (LM, LJW, MAS), experience working in Aboriginal and Torres Strait Islander health (LM, LJW, EP) and extensive experience in cervical cancer prevention (all authors).

### Governance

Cultural governance is monitored by Aboriginal and Torres Strait Islander Reference Group, Thiitu Tharrmay, at the Australian National University. Thiitu Tharrmay in Ngiyampaa language translates as 'to share/exchange knowledge'. An Aboriginal and Torres Strait Islander Project Caucus comprised of Aboriginal and Torres Strait Islander Chief and Associate Investigators provided over-arching project governance to ensure that the project is culturally safe and meets the needs and priorities of Aboriginal and Torres Strait Islander people and communities.

### Ethics approval

Ethics approval was obtained from the Australian Institute of Aboriginal and Torres Strait Islander Studies Research Ethics Committee (REC-0209) and ACON (RERC 202404), and was ratified by The Australian National University (H/2023/1434) and the University of Melbourne (2023-28454-47328-1) Human Research Ethics Committees. Individual consent was obtained from all study participants.

### Survey development

An electronic survey was designed containing questions about awareness and experience of, and future preferences for, self-collection cervical screening. The survey was designed to capture sociodemographic variables (including Indigenous status, age, sex, gender, sexuality, highest level of education, and postcode) and comprehensive feedback on

cervical screening practices and preferences. For respondents who had recently screened (i.e., since July 2022), questions asked about their experience at their most recent screen, including if they were offered the choice of self-collection, how they chose to screen, and were they given sufficient information to support their choice. Questions were primarily multiple-choice questions to ensure ease of response and consistency in data collection; however, free text fields were included to capture any specific reasons or additional comments. Questions were informed by the COM-B system for behaviour change [19] and Theoretical Domains Framework [20]. The participant information and consent forms and survey were pilot tested by both non-Indigenous and Indigenous peoples, including a non-Indigenous consumer advisory group. The survey was hosted on Qualtrics and took approximately 10–15 minutes to complete.

### Recruitment

Participants were recruited to the study through multiple channels, including a paid Meta campaign via the Cancer Council Australia Instagram and Facebook page, targeted promotion through peak Aboriginal and Torres Strait Islander Community Organisations, and through local stakeholder groups, social media, websites, newsletters, and hard-copy flyers. Recruitment was open between December 2023 and April 2024. Participants accessed the survey through an anonymous link and received a plain language statement. Informed consent was obtained from all participants. Participants were not remunerated for their time but at the completion of the survey, participants were invited to enter a draw to win a voucher valued at AUD$100. To maintain confidentiality, one study team member (CJ) managed a separate database for the voucher draw, ensuring that survey responses remained independent from participant contact details; 12 vouchers were available exclusively for Aboriginal and Torres Strait Islander participants. Random selection function was used in Excel for the prize draw. The prize draw was completed within 1 month from the final dataset being downloaded, and participant contact details were deleted following this.

### Participants

In total, 11,311 people consented to the survey, of which 677 identified as Aboriginal and/or Torres Strait Islander. Of these, 2 people were aged <24 or >74, 19 whose sex was not recorded female at birth, and 21 who answered no/unsure to having a cervix were ineligible and therefore excluded. A review of data quality, informed by published studies [21,22] and data available to the investigators from the Qualtrics platform, for identifying potentially fraudulent responses was undertaken. We excluded surveys less than 40% complete (i.e., did not reach key questions on outcome variables) or completed in <120 seconds. We then undertook manual review of surveys utilising Qualtrics fraud detection data; ReCaptca score of <0.5 and Fraud ID > 30, and groups of >5 survey which were submitted within a 1-minute window, which identified 191 responses. The manual process involved review of key free text fields (where did you hear about this survey [asked twice], qualitative response questions, and email address), some of which were specifically included to help identify fraudulent responses. Two co-authors undertook the manual process, and 80 responses were excluded on consensus (EP and CJ). This analysis reports on the 555 Aboriginal and Torres Strait Islander people who were eligible for the study. Results for non-Indigenous participants will be reported separately. This process is outlined in Fig 1.

### Data analysis

Data analysis was conducted according to Indigenous data sovereignty principles [23] and directed by previously established Aboriginal and Torres Strait Islander Governance processes at Yardhura Walani. The analysis was grounded within Rigney's Indigenist research approach [24]. This means ensuring that Aboriginal and Torres Strait Islander peoples lead, shape, and benefit from how data are collected, interpreted, and used. Data remains the property of participants, with the research team as its custodians.

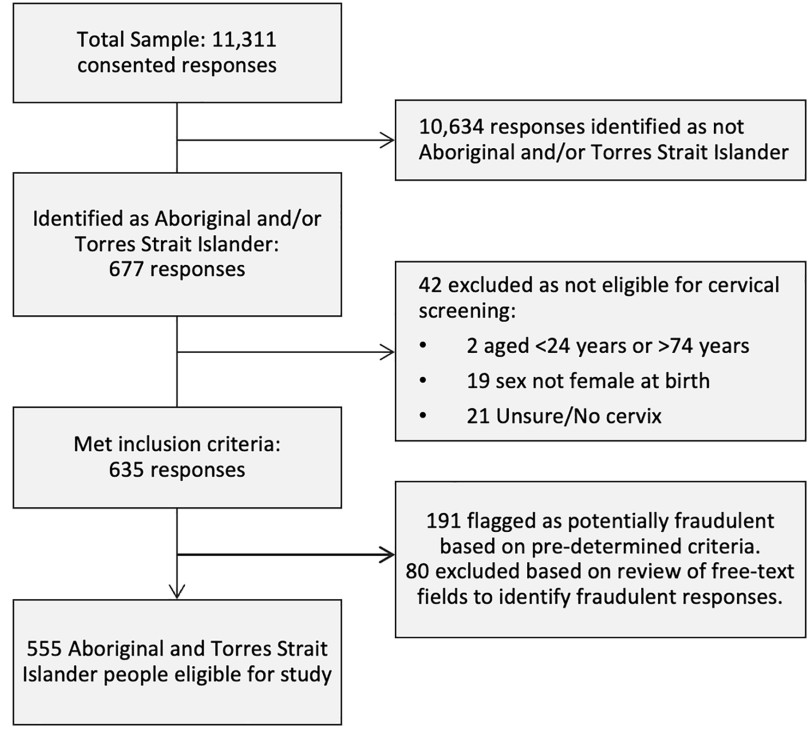

**Fig 1. Flow chart for determining survey participants.**

Research questions included: were participants offered a choice of collection methods at their most recent screen? If so, what method did they choose and why? If they chose to complete their own self-collection, did they feel they had sufficient information to support their decision?

Participants were categorised as recently screened if they responded "Within the last year (on or after July 2022)" (i.e., after the policy change) to the question "Do you remember when your last cervical screening test was?" (n = 261); previously screened if they responded "More than a year ago (before July 2022)" (n = 215), or never screened (n = 79) if they had never had a screen (n = 17), never heard of cervical screen (n = 1) or could not remember participating in cervical screening (n = 61). Postcode was used to classify the participant's place of residence by geographic remoteness using Australian Statistical Geography Standard Remoteness Structure derived from Accessibility/Remoteness Index of Australia Plus (ARIA+) [25] grouped into categories of Major City, Regional (comprising Inner Regional, Outer Regional) and Remote (comprising Remote, and Very Remote) due to participant distribution, and area-level relative disadvantage using Socio-Economic Indexes for Areas (SEIFA) Index of Relative Socioeconomic Disadvantage 2021 [26]. Summary data are provided as number of observations (n) and percentage (%). Data was analysed in STATA v18 (StataCorp, Texas, USA).

## Results

### Participants

A total of 555 screen eligible Aboriginal and Torres Strait Islander women and people with a cervix were included in the analysis. Participants predominantly identified as Aboriginal (87%), female (99%) and heterosexual (90%; Table 1). Almost half of the participants were aged 25–34 years of age (n = 239, 44%), had tertiary qualifications (n = 254, 50%), and lived in a major city (n = 224, 56%). The majority of survey participants had screened previously, either before (n = 215, 39%) or

**Table 1. Participant characteristics.**

| | Recently screened (after July 2022) | Previously screened (before July 2022) | Never screened | Total |
|---|---|---|---|---|
| **Number (%)** | **261 (47%)** | **215 (39%)** | **79 (14%)** | **555 (100%)** |
| **Indigenous Status** | | | | |
| Aboriginal | 240 (92%) | 184 (86%) | 57 (72%) | 481 (87%) |
| Torres Strait Islander | 10 (4%) | 21 (10%) | 15 (19%) | 46 (8%) |
| Both Aboriginal and Torres Strait Islander | 11 (4%) | 10 (5%) | 7 (9%) | 28 (5%) |
| **Age (years)** | | | | |
| 24 | 6 (2%) | 3 (1%) | 2 (3%) | 11 (2%) |
| 25-34 | 131 (50%) | 80 (37%) | 28 (35%) | 239 (43%) |
| 35-44 | 65 (25%) | 62 (29%) | 27 (34%) | 154 (28%) |
| 45-74 | 59 (23%) | 70 (33%) | 22 (28%) | 151 (27%) |
| **Gender** | | | | |
| Female | 261 (100%) | 211 (98%) | 77 (97%) | 549 (99%) |
| Non-Binary | 0 | <5 | <5 | 6 (1%) |
| **Intersex** | | | | |
| No | 251 (97%) | 204 (98%) | 72 (94%) | 527 (97%) |
| Yes | 9 (3%) | <5 | <5 | 18 (3%) |
| *Missing* | *1* | *7* | *2* | *10* |
| **Sexuality** | | | | |
| Heterosexual | 239 (92%) | 190 (88%) | 67 (87%) | 496 (90%) |
| Gay/Lesbian | 7 (3%) | 8 (4%) | <5 | 16 (3%) |
| Bisexual/Pansexual | 14 (5%) | 16 (7%) | 8 (10%) | 38 (7%) |
| Other | <5 | <5 | <5 | <5 |
| *Missing* | *0* | *0* | *2* | *10* |
| **Highest Education** | | | | |
| School (Primary or High) | 34 (14%) | 36 (18%) | 6 (9%) | 76 (15%) |
| Tafe/Certificate | 95 (39%) | 57 (29%) | 31 (44%) | 183 (36%) |
| Tertiary | 116 (47%) | 105 (53%) | 33 (47%) | 254 (50%) |
| *Missing* | *16* | *17* | *9* | *42* |
| **Language spoken at home** | | | | |
| English | 229 (88%) | 156 (73%) | 66 (84%) | 451 (81%) |
| Aboriginal languages | 30 (11%) | 47 (22%) | 8 (10%) | 85 (15%) |
| Torres Strait Islander languages | <5 | 10 (5%) | 5 (6%) | 17 (3%) |
| Other | 0 | <5 | 0 | <5 |
| *Missing* | *0* | *1* | *0* | *1* |
| **Duration in Australia** | | | | |
| Born Australia | 256 (98%) | 213 (99%) | 73 (92%) | 542 (98%) |
| Born Overseas | <5 | <5 | 6 (8%) | 12 (2%) |
| *Missing* | *1* | *0* | *0* | *1* |
| **Remoteness** | | | | |
| Major City | 107 (55%) | 90 (59%) | 27 (52%) | 224 (56%) |
| Regional (Inner/Outer) | 74 (38%) | 58 (38%) | 22 (42%) | 154 (39%) |
| Remote (Remote/Very Remote) | 13 (7%) | 5 (3%) | 3 (6%) | 21 (5%) |
| *Missing* | *67* | *62* | *27* | *156* |
| **Socioeconomic Status** | | | | |
| 1 Lowest SES quintile | 28 (17%) | 20 (15%) | 12 (30%) | 60 (18%) |

*(Continued)*

**Table 1.** (Continued)

| | Recently screened | Previously screened | Never screened | Total |
| | (after July 2022) | (before July 2022) | | |
|---|---|---|---|---|
| 2 | 63 (38%) | 39 (30%) | 11 (28%) | 113 (34%) |
| 3 | 37 (22%) | 31 (24%) | 9 (22%) | 77 (23%) |
| 4 | 28 (17%) | 23 (18%) | 6 (15%) | 57 (17%) |
| 5 Highest SES Quintile | 11 (7%) | 17 (13%) | <5 | 30 (9%) |
| *Missing* | *94* | *85* | *39* | *218* |

SES: socioeconomic status as determined by Socio-Economic Indexes for Areas Disadvantage Index; Remoteness as defined by Australian Statistical Geography Standard Remoteness Structure derived from Accessibility/Remoteness Index of Australia Plus (ARIA+) grouped into Major City, Regional and Remote.

after (n = 261, 47%) the policy change to offer universal access to self-collection. A smaller proportion of participants (14%, n = 79) had never screened. When asked to select from reasons for not screening, participants most commonly indicated being scared (n = 9, 11%), feeling embarrassed or shame (n = 7, 9%), or not wanting to be touched by a healthcare provider (n = 7, 9%).

## Screening practices

Among participants who had previously participated in screening, 42% (n = 201) indicated that they usually screened around the time they were due, approximately one-third (n = 158, 33%) said sometimes left it a bit too long between screens, and 25% (n = 115) reported having had one previous screen (Table 2). Of the 115 who had only screened once, 5 were aged 24 years and 28 were aged 25–29 years and therefore up to date. Participants most commonly accessed cervical screening at a doctor's clinic, or another clinic setting such as family planning or a women's clinic (Table 2). Around 13% of participants indicated usually screening at an Aboriginal Community Controlled Health Organisation or Aboriginal Medical Service. When asked what prompted participants to get a cervical screening test, the most frequently identified reason was having gone to the healthcare provider for another reason and being reminded (n = 130, 27%), highlighting the importance of opportunistic screening. Other prompts included receiving a reminder (such as a letter or SMS) from their usual clinic (n = 110, 23%), or remembering they were due for screening (n = 109, 23%). Reminder based prompts were frequently cited for those who screened around when due, with 33% (n = 66) citing a reminder from their usual clinic and 27% (n = 55) remembering they were due. In contrast, participants who reported 'leaving it a bit too long' or had only screened once most frequently cited being reminded by their healthcare provider during a visit for another issue (n = 51, 32% and n = 30, 26%). Many also reported screening because they were worried about a symptom or health problem (n = 95, 20%). This was the most frequently cited reason for screening among those who had only screened once (n = 39, 34%).

## Experiences of recently screened participants

Participants who reported screening since the policy change recommending universal access to the option of self-collection were asked additional questions about their most recent experience. In total, 46% (n = 118) of recently screened participants, used HPV self-collection, either doing it themselves at home (n = 41, 35%) or at the clinic (n = 48, 41%), or assisted by a healthcare provider without a speculum (n = 29, 24%; Table 3). Over half of recently screened participants reported having a clinician collected their sample with a speculum (n = 133, 51%).

Among recently screened participants who described screening 'around when due', almost two-thirds reported using clinician-collection with a speculum at their last screen (n = 87, 65%), while 34% used self-collection (n = 46;

**Table 2. Screening practices of participants that have previously screened to the policy change offering universal access to self-collection (i.e., before or after July 2022).**

| Screening practices | When was last cervical screen | | Total |
|---|---|---|---|
| | After July 22 | Before July 22 | |
| **Total (%)** | 261 (55%) | 215 (45%) | 476 (100%) |
| **How often do you screen** | | | |
| Around when due | 135 (52%) | 66 (31%) | 201 (42%) |
| Leave it a bit long | 81 (31%) | 77 (36%) | 158 (33%) |
| Only had one* | 44 (17%) | 71 (33%) | 115 (24%) |
| *Missing* | 1 | 1 | 2 |
| **Usual screening location** | | | |
| Doctors clinic | 115 (44%) | 93 (43%) | 208 (44%) |
| Women's/Sexual Health clinic | 92 (35%) | 84 (39%) | 176 (37%) |
| ACCHO/AMS | 36 (14%) | 24 (11%) | 60 (13%) |
| Gynaecologist | 17 (7%) | 11 (5%) | 28 (6%) |
| Other | 0 | <5 | <5 |
| *Missing* | 1 | 1 | 2 |
| **Why did you get your last CST? (select all that apply)^** | | | |
| I went to the healthcare provider for another reason and they reminded me | 81 (31%) | 49 (23%) | 130 (27%) |
| A reminder letter, email, SMS (or similar) from my usual clinic | 56 (21%) | 54 (25%) | 110 (23%) |
| I remembered I was due for a screening test | 66 (25%) | 43 (20%) | 109 (23%) |
| I was worried about a symptom or health problem | 40 (15%) | 55 (26%) | 95 (20%) |
| A reminder letter from the National Cancer Screening Register/Pap Test register | 50 (19%) | 37 (17%) | 87 (18%) |
| The health worker told me I was due | 36 (14%) | 37 (17%) | 73 (15%) |
| I wanted to follow-up after a previous abnormal cervical screen | 34 (13%) | 25 (12%) | 59 (12%) |
| I saw something in the media (television, radio, social media) | 14 (5%) | 6 (3%) | 20 (4%) |
| Other | 22 (8%) | 9 (4%) | 31 (7%) |

CST: cervical screening test. HCP: healthcare provider, ACCHO/AMS: Aboriginal Community Controlled Health Organisation (ACCHO) or Aboriginal Medical Service (AMS) *Of those screened only once, 5 were aged 24 years and 28 were aged 25–29 years and therefore up to date. ^Note: Participants could choose multiple reasons therefore percentages will not add up to 100%.

Table 3). In contrast, self-collection was more common among those who 'left it a bit long' (n = 47, 59%) or had only one previous screen (n = 25, 57%). Clinician-collection was reported by 40% (n = 32) and 32% (n = 14) in these groups respectively (Table 3).

Over half of respondents (n = 151, 58%) recalled being offered the choice of self-collection at their last screen (Table 3). Of those offered the choice, 67% chose to screen using self-collection (n = 101). A further 15 participants were not offered a choice and 2 who could not recall whether or not they had been offered a choice screened using self-collection.

The majority of participants who screened using clinician-collection (61%, n = 81) were not offered a choice of collection method. However 50 participants were offered a choice and elected to have a clinician-collected sample, and an additional two couldn't remember.

## Reasons for collection method

Among the 89 participants who collected their own sample either at home or in the clinic, the most commonly reported reasons for so, whether offered a choice or not, were because it was less embarrassing (n = 49, 55%), they felt in control of their body (n = 34, 38%), and it was less scary (n = 27, 30%; Table 4). Nearly half (n = 43, 49%) said they would have

 

**Table 3. Collection method of participants recently screened since the policy change offering universal access to self-collection (i.e., after July 2022), by reported screening frequency and whether choice was offered.**

| | Self-collection | | | | HCP with speculum (clinician-collection) | Unsure | Total |
|---|---|---|---|---|---|---|---|
| | In the Clinic | At home or elsewhere | HCP assisted without speculum | Total self-collection | | | |
| **Total** | **48 (19%)** | **41 (16%)** | **29 (11%)** | **118 (46%)** | **133 (51%)** | **8 (3%)** | **259* (100%)** |
| *By screening frequency* | | | | | | | |
| Around when due | 28 (21%) | 10 (7%) | 8 (6%) | 46 (34%) | 87 (65%) | <5 | 134 (100%) |
| Leave it a bit long | 16 (20%) | 22 (28%) | 9 (11%) | 47 (59%) | 32 (40%) | <5 | 80 (100%) |
| Only one screen | <5 | 9 (20%) | 12 (27%) | 25 (57%) | 14 (32%) | 5 (11%) | 44 (100%) |
| *Whether choice was offered* | | | | | | | |
| Offered choice | 42 (28%) | 33 (22%) | 26 (17%) | 101 (67%) | 50 (33%) | 0 | 151 (100%) |
| Not offered choice | 6 (6%) | 6 (6%) | <5 | 15 (15%) | 81 (81%) | <5 | 100 (100%) |
| Can't remember | 0 | <5 | 0 | <5 | <5 | <5 | 8 (100%) |

HCP: healthcare provider. *Responses from 2 participants screened after July 2022 missing.

screened anyway if self-collection were not an option. However, 31% (n = 27) indicated they might have screened, and 20% (n = 18) reported they would not have screened without self-collection, highlighting the importance in supporting participation.

Among those who chose self-collection assisted by a healthcare provider (i.e., without a speculum), the most commonly reported reasons for doing so was they wanted the health professional to have a look without a speculum and ensure everything was OK (n = 19, 66%) or they believed the provider would collect a better sample (n = 14, 48%).

For those who screened using a clinician-collected sample, the most commonly reported reasons included always having had it done by a healthcare provider (n = 66, 50%), wanting the healthcare provider to have a look (n = 25, 19%), or believing the healthcare provider would collect a better sample (n = 17; 13%; Table 4). Of those who were not offered a choice who screened using clinician-collection, 40 were not told about self-collection (31%); Table 4).

### Information to support the choice of self-collection

Participants who recently screened and collected their own sample at home or in the clinic (n = 89) were asked if they were given information to support their choice.

More than half of these participants recalled being provided the specific pieces of information needed to make an informed decision around self-collection, such as needing to return for a clinician-collected sample if HPV was found, that they could ask for assistance to collect the sample, or how they would receive test results (Fig 2). Only 55% of participants felt they had enough information to choose between self-collection and clinician-collection methods.

When asked about their agreement with statements regarding self-collection, most participants agreed that they understood how to collect their sample (70% agreement), that the test was accurate (69% agreement) and that self-collection was easy to do (66% agreement). Participants were less confident in their ability to correctly collect the sample (55% agreement), and 33% were worried having to return for a follow-up test (Fig 3).

### Discussion

To our knowledge, this is the first study to examine the experience of cervical screening among Aboriginal and Torres Strait Islander women and people with a cervix since the introduction of universal access to self-collection in July 2022.

**Table 4. Reason for collection method among recently screened participants (i.e., after July 2022), by whether choice was offered.**

| | Offered choice at last cervical screen | | | Total |
|---|---|---|---|---|
| | **Offered choice** | **Not offered choice** | **Can't remember** | |
| **Recently screened participants (since July 2022)** | 151 (58%) | 100 (39%) | 8 (3%) | 261 (100%)* |
| **Collected own self-collected sample at home or in clinic (select all that apply)^** | 75 (84%) | 12 (13%) | <5 | 89 (100%) |
| It is less embarrassing | 40 (82%) | 8 (16%) | <5 | 49 (100%) |
| It helps me feel in control of my body | 28 (82%) | 5 (15%) | <5 | 34 (100%) |
| It is not as scary | 23 (85%) | <5 | <5 | 27 (100%) |
| It is more convenient | 18 (75%) | 5 (21%) | <5 | 24 (100%) |
| It is less painful | 19 (83%) | <5 | 0 | 23 (100%) |
| My HCP suggested it as an option | 13 (81%) | <5 | 0 | 16 (100%) |
| A friend/family member had a good experience with self-collection | 12 (86%) | <5 | <5 | 14 (100%) |
| It is just as accurate as a sample collected by a HCP | 11 (79%) | <5 | <5 | 14 (100%) |
| I had a bad experience in the past when a sample was collected by a HCP using a speculum | 7 (64%) | <5 | <5 | 11 (100%) |
| The information I received made it look like a good option | 7 (88%) | <5 | 0 | 8 (100%) |
| Other | 5 (63%) | <5 | 0 | 8 (100%) |
| **Used HCP assisted self-collection (i.e., no speculum) (select all that apply)^** | 26 (90%) | <5 | 0 | 29 (100%) |
| I want my HCP to have a look, even without a speculum, and make sure that everything is ok | 17 (89%) | <5 | 0 | 19 (100%) |
| I thought that the HCP could collect a better sample | 13 (93%) | <5 | 0 | 14 (100%) |
| It's less embarrassing | <5 | 0 | 0 | <5 |
| I cannot reach, or it is not comfortable for me to reach | <5 | 0 | 0 | <5 |
| I do not like to touch myself there | <5 | 0 | 0 | <5 |
| **Used clinician-collected sample (i.e., with a speculum) (select all the apply)^** | 50 (38%) | 81 (61%) | <5 | 133 (100%) |
| I was not told about the option to use self-collection | <5 | 40 (98%) | 0 | 41 (100%) |
| I have always had it done by a HCP | 25 (38%) | 39 (59%) | <5 | 66 (100%) |
| I wanted my HCP to have a look | 13 (52%) | 12 (48%) | 0 | 25 (100%) |
| I did not think I would take the self-collected sample properly | 11 (65%) | 6 (35%) | 0 | 17 (100%) |
| I did not think self-collection was accurate | 6 (60%) | <5 | 0 | 10 (100%) |
| I am not eligible for self-collection | <5 | 7 (88%) | 0 | 8 (100%) |
| I thought I would need to return for another test anyway if I used self-collection | <5 | <5 | 0 | 5 (100%) |
| I do not know why | <5 | <5 | 0 | 5 (100%) |
| I needed more information about self-collection | <5 | <5 | 0 | <5 |
| My HCP told me they did not offer self-collection | <5 | 0 | 0 | <5 |
| Other | 6 (33%) | 12 (67%) | 0 | 18 (100%) |

HCP: healthcare provider. **Responses from 2 participants screened after July 2022 are missing and 8 were unsure of collection method. ^Note: Participants could choose multiple reasons for their chosen collection method therefore percentages will not add up to 100%.

This change means that anyone who is eligible for cervical screening should be offered the choice of HPV testing on clinician-collected cervical sample or on a self-collected vaginal sample [27]. Our survey was conducted between December 2023 and April 2024, almost 18 months after the policy shift. Among survey participants screened in this approximately 18-month period, 39% of participants reported they were not offered a choice. This is consistent with reports of implementation challenges experienced prior to universal access, including limited promotion, and lack of confidence in test accuracy, leading to low self-collection uptake [10,28]. The policy change to universal access, alongside improved provider education on updated clinical guidelines and more accredited pathology services had already driven a steady increase in use of self-collected tests prior to this study, from 1.2% of all tests immediately prior to universal access to 23.9% in Q3

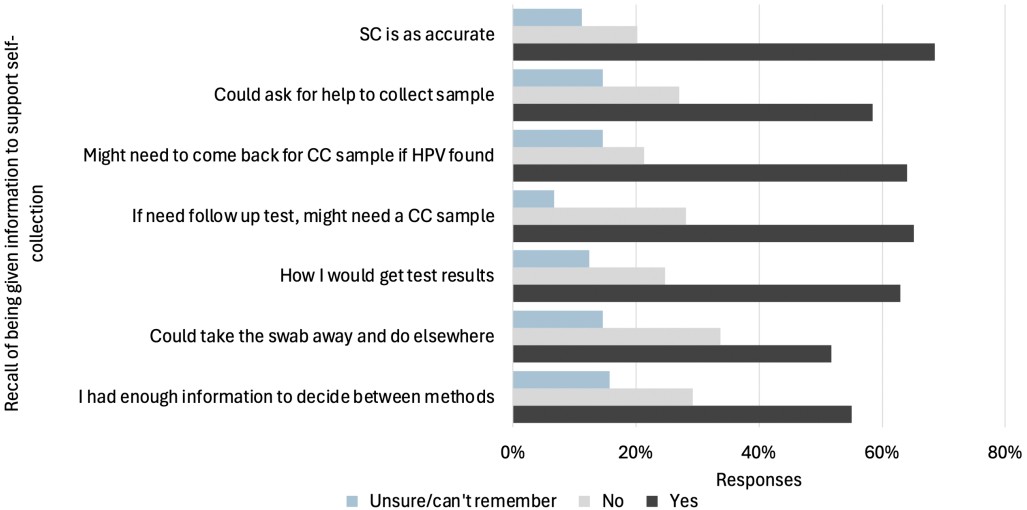

**Fig 2. Recall of being given information to support informed decision making among those who recently screened using self-collection at home or in the clinic (n = 89).** SC: self-collection, CC: clinician collection, HCP: healthcare provider, HPV: Human Papillomavirus.

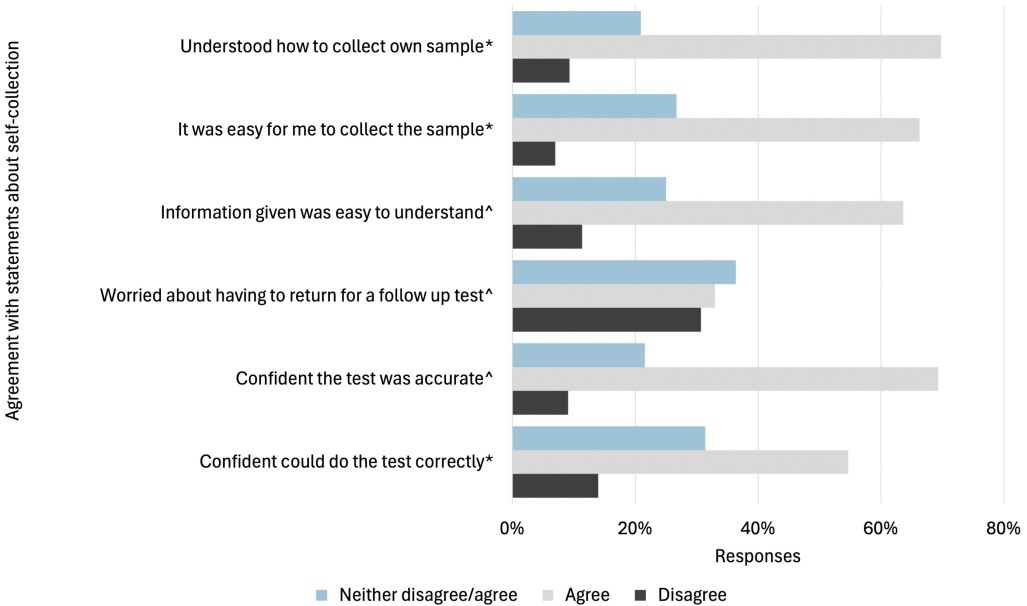

**Fig 3. Agreement with statements among those who recently who recently screened using self-collection (n = 89).** *Responses from 86 participants; ^ Responses from 88 participants.

2023 [29]. This has continued to increase since then, reaching 45.96% of all tests in Q2 2025, after the commencement of a major national campaign in September 2024 that promoted awareness of the option to use self-collection and which was preceded by a range of promotion activities encouraging providers to be ready to offer self-collection [29,30].

Our findings indicate a higher proportion of self-collected samples than national population-wide statistics at the time of the survey with 46% of recently screened participants reporting using self-collection either in the clinic, at home, or

assisted by a healthcare provider. Our findings also show that when offered a choice, 67% of participants chose self-collection. Participants selected this option as it was less embarrassing, scary and painful, let them feel in control of their body, and was more convenient. These findings are consistent with existing literature where self-collection is shown to promote privacy, bodily autonomy and control among Indigenous women and people with a cervix, in particular among those who would not normally participate in screening [16,18,31,32]. Interestingly, the majority of this group had previously screened and over 40% indicated they typically screened on time. Our results confirm that having a more positive screening experience by avoiding trauma, pain, discomfort and embarrassment are important factors for women and people with a cervix, even among those who typically screen. This is particularly important to consider for under- or never screened participants who indicated they chose not to screen because they were scared, felt embarrassed or shame, didn't want to be touched by a healthcare provider. Research shows that cervical screening should be conducted in a way that empowers Indigenous women to feel in control of their bodies, health, and health decision-making [33] and that the choice of self-collection for under- and never-screened Aboriginal and Torres Strait Islander women and people with a cervix is integral to participation [16–18,31]. Our research demonstrated that participants who screened "around when due" predominantly chose clinician-collection with a speculum suggesting comfort or familiarity with the traditional method. However, 34% reported "switching" to self-collection – consistent with national figures of 38% of people who screened switching from clinician-collection to self-collection [29]. In contrast, those who "left it a bit long" were more likely to opt for self-collection or assisted self-collection without a speculum, indicating a preference for less invasive or more accessible options. This underscores the critical role of self-collection in enabling participation, particularly for those who may otherwise delay or avoid screening. Among those who had only one screen since the policy change, screening methods were more evenly distributed, reflecting varied uptake and possibly differing levels of awareness or access. Ensuring a positive first screening experience is a key factor in supporting ongoing participation in cervical screening programs [31].

When asked about the information provided to support their decision to screen using self-collection, only 55% felt they had enough information to choose between collection methods. It is important to note that these responses occurred in a group of participants where almost half had tertiary qualifications and almost all were English-speaking. This highlights the role of healthcare providers in not only providing information but supporting clients to make an informed decision, and more broadly to the need to embed health literacy into systems and environments to support screening participants to access, understand and act upon health-related information [34]. This is particularly important to consider in the case of cervical screening to support decision making around any follow-up tests that may be required. For example, almost half of our survey participants were aged 25−35 years of age and national statistics show relatively high rates of oncogenic HPV (not 16/18) detected in these age groups, particularly among Aboriginal and Torres Strait Islander women and people with a cervix [5]. Guidelines reinforce that to support informed choice, any conversation with these participants should involve discussing the likelihood of follow-up tests, including explaining that those who test positive for non-16/18 HPV types on a self-collected sample may need to return for a clinician-collected sample for cytology [27]. This information is important to ensure informed decision making around the collection method.

To our knowledge this is the first insight into the preferences, use and experiences of Aboriginal and Torres Strait Islander women and people with a cervix in relation to self-collection assisted by a healthcare provider (i.e., without a speculum). The most common reasons for doing so wanting an examination and a perception that the healthcare provider would take a better sample. Aboriginal and Torres Strait Islander women have raised concerns in previous research that a clinician-collected test was more accurate and professional, and had expressed concerns around self-sampling incorrectly or hurting themselves [18]. This reinforces the need to educate clients about the accuracy of self-collection and the need for only a low to mid vaginal swab [12,13]. Self-collection with the help of a trusted healthcare provider also offers an alternative for those who struggle with self-collection. While less than five participants in our sample indicated being unable to collect their sample as a reason for choosing this method, this possibly reflects the younger cohort with potentially fewer physical limitations. However, for women and people with a cervix who have physical limitations that may hinder their

ability to self-collect, it is important that their healthcare provider offers them this choice and supports them in making the best decision for their individual needs.

In Australia, self-collection is offered via a practitioner-supported model of care. Our survey results support the role of the healthcare provider in the delivery of cervical screening, with most participants screening opportunistically due to being reminded by their healthcare provider, or in response to a reminder from their clinic. Maintaining the role of the healthcare provider, rather than mail-out models seen in other high-income countries, also helps address any perceived barriers to self-collection, such as concerns about collecting the sample correctly or its accuracy. Additionally, it ensures that follow-up care is appropriately managed if the test results are positive for HPV. This model maintains the critical role of healthcare providers in delivering comprehensive care and supports informed decision-making [35]. Over half of participants lived in major cities, where access to primary care is generally easier. Despite this, participants in urban areas still face barriers such as cost, transport, caring and work commitments, and limited access to culturally safe care. These factors can significantly impact participation in cervical screening. For those in regional or remote areas, access is likely even more restricted, and alternative models may be needed to improve cervical screening uptake.

Just over half of recently screened participants reported having a clinician-collected sample (i.e., with a speculum) at their most recent cervical screening test. For some people under surveillance or those exhibiting symptoms a co-test is indicated, necessitating a clinician-collected sample. This fits with a small proportion of participants (6%) being told they were not eligible for self-collection. While the majority of participants who were screened using clinician-collected samples were not offered a choice of collection method (61%), some who were offered a choice elected to have a clinician-collected sample. This highlights the importance of offering screening participants a choice of collection methods, ensuring they are fully informed and supported in making the decision that best suits their needs. When asked why, participants indicated they chose a clinician-collected sample because they had always had it done by a healthcare provider, they wanted the healthcare provider to have a look, or they thought their healthcare provider would be able to collect a better sample. Providers can reassure their clients that genital inspection is not necessary in the absence of vulvovaginal symptoms or risk factors for vulvar disease [27]. This reinforces the importance of empowering eligible clients to make an informed decision around collection methods. In doing so, this also honours the sovereign rights of Aboriginal and Torres Strait Islander people to govern health decisions.

Our findings – that is, that Aboriginal and Torres Strait Islander screening participants value the choice of collection methods and require support to make an informed decision – are potentially even more important for these groups who likely need greater information and support. These findings provide an important benchmark for a planned future survey using similar methodologies, that will be able to explore whether there have been changes as self-collection becomes more established practice and following the major national campaign. The 'Own it!' campaign was developed in partnership with the National Aboriginal Community Controlled Health Organisation (NACCHO) and features Aboriginal and Torres Strait people who are a primary audience for the campaign, targeting both provider and participant focused education and awareness.

## Strengths and limitations

This survey successfully reached potential participants through a targeted social media recruitment campaign alongside traditional recruitment methods. Online surveys are inherently at risk of fraudulent responses by bots or individuals aiming to complete multiple survey responses to receive prize incentives. Our recruitment method aimed to minimise the risk of attracting bots using a paid meta campaign which specifically targeted Instagram and Facebook users based on age and geographical location and not mentioning the prize draw in the caption text of any social media posts. In our study, we implemented multiple layers of protection in our survey, including captchas and honeypot questions (imbedded in the survey using JavaScript, and only visible to bot responses), and undertook a method of data review to ensure data quality. Based on our experience of recruiting a large-scale online survey, a multi-pronged approach is required to control for fraudulent survey responses and maintain data integrity.

A key limitation of the study was that it was conducted exclusively online, which may have introduced bias by excluding individuals without reliable access to the internet or a computer. The sample has a higher proportion of participants aged 25–34 years compared to the total sampling frame (44% vs 35%), and fewer participants aged 65–74 years (1% vs 7%). Similarly, the sample over-represents those living in Major Cities (56% vs 41%) and under-represents those living in Very Remote areas (1% vs 11%) compared to the total sampling frame. This likely reflects the primary method used to recruit to the survey (online survey with advertisement via a Meta Campaign). Additional research using alternative methods is needed to engage with and understand the experience among populations in very remote areas or those experiencing digital exclusion.

Another limitation of the data is that participants' cervical screening history was self-reported and collected from a non-probabilistic, convenience sample, meaning it may not be representative and cannot be verified against clinical records. This is particularly of note for individuals whose screening occurred several years ago, introducing differential recall bias potentially affecting the reliability of self-reported data. To strengthen the reliability of findings, program-level data on Aboriginal and Torres Strait Islander participation in cervical screening, with details on collection methods, is needed to understand the uptake nationally and support accurate monitoring of cervical screening.

Finally, this study did not seek to capture qualitative data exploring participants' experiences with different screening methods. Additional research is required to understand the personal, cultural, or emotional factors that may influence choice.

## Conclusions

This study indicates a preference for self-collection among Aboriginal and Torres Strait Islander women and people with a cervix when offered a choice between clinician collected and self-collection. Findings also highlight the vital role of health-care providers in the delivery of cervical screening, in particular supporting clients to confidently choose and complete self-collection. There is a significant opportunity to ensure all screen-eligible participants are aware of self-collection, are offered this option and can make an informed decision to participate in cervical screening, due to the national promotional campaign running during 2024–2025.

## Supporting information

**S1 File. Inclusivity in global research questionaire.**
(DOCX)

**S2 File. Survey.**
(PDF)

## Acknowledgments

We wish to thank those who participated in the study. We acknowledge the assistance and guidance of the Screen Your Way Investigators, Aboriginal and Torres Strait Islander Caucus as well as Aboriginal and Torres Strait Islander reference group Thiitu Tharrmay. Professor Lisa J Whop (Wagadagam, Gumulgal) from Yardhura Walani, Australian National University, (Lisa.Whop@anu.edu.au) is Principal Investigator of Screen Your Way. Chief Investigators are Associate Professor Julia Brotherton from the Centre for Health Policy, Melbourne School of Population and Global Health, University of Melbourne; Professor Gail Garvey (Kamilaroi) from the First Nations Cancer & Wellbeing Research Program, The University of Queensland; Dr Tamara Butler (Undumbi) from both the Yardhura Walani, Australian National University and the First Nations Cancer & Wellbeing Research Program, The University of Queensland; Associate Professor Mark Wenitong (Kabi Kabi) from the School of Public Health, The University of Queensland; Associate Professor Megan Smith from the Sydney School of Public Health, The University of Sydney (Named Author); Associate Professor Claire Nightingale from

the Centre for Health Policy, University of Melbourne (Named Author); Professor Marion Saville from the Australian Centre for Prevention of Cervical Cancer; Professor Rebecca Guy from the Kirby Institute, UNSW Sydney; and Professor Joan Cunningham from the Menzies School of Health Research, Charles Darwin University.

Associate Investigators include Claudette "Sissy" Tyson (Kuku Yalanji) from the Queensland Centre of Excellence in Aboriginal and Torres Strait Islander Primary Health Care; Sonya Egert (Noonuccal Goenpul) from Inala Wangarra; Kristine Falzon (Gummeah, WandiWandian, Wodi Wodi, Walbunja) from Waminda South Coast Women's Health and Wellbeing Aboriginal Corporation; Professor Bev Lawton (nō Ngāti Porou) from Te Tātai Hauora o Hine-National Centre for Women's Health Research Aotearoa, Victoria University of Wellington; Professor Karen Canfell from the Sydney School of Public Health, The University of Sydney; Associate Professor Natalie Taylor from the School of Population Health, UNSW Sydney; and Dr Hamish McManus from the Kirby Institute, UNSW Sydney. We also thank Josephine Mondino, Cancer Council Australia, for providing support and advice for the Meta recruitment campaign. Ownership of Aboriginal and Torres Strait Islander knowledges and cultural heritage is retained by the informants.

## Author contributions

**Conceptualization:** Louise E. Mitchell, Emily Phillips, Chloe J. Jennett, Claire Bavor, Tessa Saunders, Claire Nightingale, Megan A. Smith, Lisa J. Whop.

**Data curation:** Louise E. Mitchell, Emily Phillips, Chloe J. Jennett, Claire Bavor, Tessa Saunders.

**Formal analysis:** Louise E. Mitchell, Chloe J. Jennett, Claire Bavor, Tessa Saunders.

**Funding acquisition:** Claire Nightingale, Megan A. Smith, Lisa J. Whop.

**Investigation:** Louise E. Mitchell, Chloe J. Jennett, Claire Bavor, Tessa Saunders, Claire Nightingale, Megan A. Smith, Lisa J. Whop.

**Methodology:** Louise E. Mitchell, Emily Phillips, Chloe J. Jennett, Claire Bavor, Tessa Saunders, Claire Nightingale, Megan A. Smith, Lisa J. Whop.

**Project administration:** Emily Phillips, Chloe J. Jennett, Claire Bavor, Tessa Saunders, Claire Nightingale, Megan A. Smith, Lisa J. Whop.

**Resources:** Tessa Saunders.

**Supervision:** Lisa J. Whop.

**Visualization:** Louise E. Mitchell.

**Writing – original draft:** Louise E. Mitchell.

**Writing – review & editing:** Louise E. Mitchell, Emily Phillips, Chloe J. Jennett, Claire Bavor, Tessa Saunders, Claire Nightingale, Megan A. Smith, Lisa J. Whop.

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
