## [Decision Letter · Decision Letter 0]

8 Sep 2025

Dear Dr. Mitchell,

Thank you for submitting your manuscript to PLOS ONE. After careful consideration, we feel that it has merit but does not fully meet PLOS ONE’s publication criteria as it currently stands. Therefore, we invite you to submit a revised version of the manuscript that addresses the points raised during the review process.

We look forward to receiving your revised manuscript.

Kind regards,

Visalini Nair-Shalliker

Academic Editor

PLOS ONE

Journal Requirements:

4. One of the noted authors is a group or consortium Chief Investigators: Brotherton, Garvey, Butler, Wenitong, Smith, Nightingale, Saville, Guy, Cunningham; Associate Investigators: Tyson, Egert, Falzon, Lawton, Canfell, Taylor, McManu. In addition to naming the author group, please list the individual authors and affiliations within this group in the acknowledgments section of your manuscript. Please also indicate clearly a lead author for this group along with a contact email address.

5. We note that you have indicated that there are restrictions to data sharing for this study. For studies involving human research participant data or other sensitive data, we encourage authors to share de-identified or anonymized data. However, when data cannot be publicly shared for ethical reasons, we allow authors to make their data sets available upon request. For information on unacceptable data access restrictions, please see http://journals.plos.org/plosone/s/data-availability#loc-unacceptable-data-access-restrictions.

6. In the online submission form, you indicated that Individual survey responses cannot be provided as participants did not consent to the public release. However, summarised findings with suppressed counts to maintain participant privacy may be shared upon reasonable request. Requests should be directed to the corresponding author (louise.mitchell@anu.edu.au) and will require approval from the relevant Human Research Ethics Committees (HRECs) listed in the Ethics Approval section.

7. One of the noted authors is a group or consortium on behalf of Screen Your Way Investigator Team. In addition to naming the author group, please list the individual authors and affiliations within this group in the acknowledgments section of your manuscript. Please also indicate clearly a lead author for this group along with a contact email address.

Reviewers' comments:

Reviewer's Responses to Questions

**Comments to the Author**

1. Is the manuscript technically sound, and do the data support the conclusions?

Reviewer #1: Yes

Reviewer #2: Yes

2. Has the statistical analysis been performed appropriately and rigorously?

Reviewer #1: Yes

Reviewer #2: Yes

3. Have the authors made all data underlying the findings in their manuscript fully available?

Reviewer #1: No

Reviewer #2: Yes

4. Is the manuscript presented in an intelligible fashion and written in standard English?

Reviewer #1: Yes

Reviewer #2: Yes

Reviewer #1: Manuscript ID: PONE-D-25-29330

Title: Experiences of HPV self-collection among Aboriginal and Torres Strait Islander women

and people with a cervix

*Comments to the Author

This article describes a survey of Australian women and people with a cervix that asked questions about awareness and experience of and preferences for, self-collection cervical cancer screening. Australia implemented the option for self-collection in July 2022 and this study focused on the subset of the respondents who self-identified as Aboriginals and Torres Strait Islanders months after this option was implemented.

This is an important question for this population as they have historically participated less in cervical cancer screening than non-Indigenous populations due to multiple factors, with corresponding higher incidence and mortality rates.

A strength of this research study was the composition of the research team, which included members from the National Centre for Aboriginal and Torres Strait Islander Wellbeing Research, who also had lived experience. Other research team members have experience including leadership in public health in this population as well. This study received ethics approval from multiple organizations including the Australian Institute of Aboriginal and Torres Strait Islander Studies Research Ethics Committee. The survey materials were pilot tested with Indigenous people as well. Another strength was the rigorous evaluation of the data collected to minimize bias from non-human responses in an online survey.

The manuscript was easy to read and follow. This research will contribute to understanding the preferences for Indigenous women for HPV self-sampling and perhaps increase participation in this population.

General Comments

Suggest that you include the questionnaire as an appendix as that would provide a suitable resource for other Indigenous groups to consider using some or all of the questions to assess self-sampling in their populations.

Introduction

The background information provided important details on changes to the cervical cancer screening guidelines in Australia over the past 10 years. The rationale for evaluating self-sampling in the Aboriginal and Torres Strait Islander was reasonably supported but whether this change will overcome barriers to participation by these people was not as clearly established. The cited literature was relevant and generally very recent; however, there was a lack of information about HPV vaccinations uptake in this population. Please include additional information as that is likely most relevant in the 25–35-year-olds mentioned on page 15 beginning on line 14 regarding positive oncogenic HPV screens.

Materials and Methods

Page 5 line 2 – an electronic version of the survey will not be accessible to people without access to computers and the internet. How would this potentially bias the results in general but specifically this population who tend to live in more remote areas?

Page 6 line 5 – how were responses kept separate from participants enrolling in the draw for a voucher? Was only one voucher offered for all respondents to the survey?

Results

Table 1 – many of the reported counts for variable categories do not sum up to the column totals. Please include a category for missing data so that it is clear to the reader if questions were not answered. Data suppression is fine but likely not needed for missing information.

- How is remoteness defined?

Table 2 – many of the reported counts for variable categories do not sum up to the column totals. Please include a category for missing data so that it is clear to the reader if questions were not answered.

– are the categories for the question “What prompted you to get a CST “ mutually exclusive? It does not seem so. Please include information about the question in the table, for example add (select all that apply).

- Add a row total to this table as results for individual questions are reported for the combined groups in the Screening Practices section.

- Could report some differences between the two groups, such as “Around when due for screening“ and “I was worried about a symptom or health problem.” The reasons for these differences could then be explored in the Discussion.

Table 3 – add a column with the sample size for each question rather than including it with the question.

- As with Tables 1 and 2, many of the reported counts for variable categories do not sum up to the sample sizes. Please include a category for missing data so that it is clear to the reader if questions were not answered.

- are the categories for the questions “Reasons for choosing HCP assisted self-collection (i.e. no speculum “ and “Reasons for choosing self-collection” mutually exclusive? It does not seem so. Please include information about the question in the table, for example add (select all that apply). The footnote for this table is helpful but adding information to the question makes this easier to understand too.

- Why is the sample size 89 for this question “Reasons for choosing self-collection” when it seems it should be 116? Is this the number who answered this question out of the 116 respondents?

- It would be more informative to look at the responses to this question separately for the groups who were or were not offered a choice for screening: “Reasons for choosing a clinician-collected sample (i.e. with a speculum).” The reasons might be different and that could inform who to target with educational materials in the future.

Page 13 lines 3 to 13: the counts and percentages presented here do not tally with the numbers in table 3, unless these are the results only from those who answered these questions, not whether they did self-collection. For example, Screened using self-collection^ should be 116 in total not 89. Please include more information on these results (e.g., 89 out of possible 116 answered this question ….) as they are difficult to interpret and understand as they are written now.

Figure 2 – the figure legend should indicate the results are given as percentages. It should also indicate multiple responses are possible and why the results do not sum to 100%. The graph currently shows the total is 100% but it is not for the data presented. Including a missing or not answered category would improve the interpretation of these results. The figure is also quite blurry.

Results from this Table 2 can be shown better using multiple bar charts, side by side. See this web page for an example: https://communities.sas.com/t5/Graphics-Programming/Multiple-Bar-charts-side-by-side/td-p/277586

Discussion/Conclusions

Page 16 Lines 11-19- Are there challenges with access to primary care in this population due to remote locations or fewer options? Or was your sample more likely from an urban setting (see comment about study limitation below)?

Page 17 line 15 “… in a major (city)”?

Strengths and Limitations section – only mentions strengths. A major limitation is self-report information was collected from a non-probabilistic, convenience sample. The data cannot be verified using actual examination data. Another limitation is the use of an electronic version of the survey that would likely exclude people who don’t have access to a computer and the internet. Another potential limitation is differential recall bias – the people recently screened will likely have more accurate recall of their appointment compared to someone who might have been screened several years ago.

The Discussion does not describe how the results of this study could improve participation by people who have never screened or are not up to date in their screening? Does the National Campaign that started in 2024 target these groups? Or their healthcare providers? Please expand so that the knowledge gained from this survey can improve cervical cancer screening in this population, and possibly the more general screen-eligible population in Australia.

Reviewer #2: The manuscript is scientifically sound, well written and of relevance to the current Australian healthcare landscape. I would encourage the acceptance of this paper with a few ramifications. I would also like to add a bit more content regarding cultural safety of healthcare spaces.

**Do you want your identity to be public for this peer review?** For information about this choice, including consent withdrawal, please see our Privacy Policy

Reviewer #1: No

Reviewer #2: No

---

## [Author Response · Author response to Decision Letter 1]

29 Oct 2025

Thank you for your thoughtful and considered feedback on our manuscript. We appreciate the time taken to review our work. We have addressed these comments and suggestions, and attach a revised version of the manuscript for your consideration.

---

## [Decision Letter · Decision Letter 1]

29 Jan 2026

Dear Dr. Mitchell,

Thank you for submitting your manuscript to PLOS ONE. After careful consideration, we feel that it has merit but does not fully meet PLOS ONE’s publication criteria as it currently stands. Therefore, we invite you to submit a revised version of the manuscript that addresses the points raised during the review process.

We look forward to receiving your revised manuscript.

Kind regards,

Visalini Nair-Shalliker

Academic Editor

PLOS One

Journal Requirements:

Additional Editor Comments:

Dear Dr Mithcell

Thank you for submitting the revised version of your manuscript PONE-D-25-29330R1 entitled "Experiences of HPV self-collection among Aboriginal and Torres Strait Islander women and people with a cervix."to PLOS ONE. I have now completed my evaluation of your revisions and am pleased to inform you that the changes satisfactorily address the concerns raised by the reviewers.

On this basis, I am happy to provisionally accept your manuscript. It will now proceed to the next stage of our editorial workflow, which includes final checks by the journal’s staff to ensure compliance with PLOS ONE’s publication criteria, ethical standards, and formatting requirements.

Please note that during this stage you may be contacted by the journal office if any additional minor edits or clarifications are required to finalize the manuscript for publication.

Thank you once again for your careful attention to the reviewers’ comments and for choosing PLOS ONE as the venue for your work. We look forward to seeing your article move toward publication.

Kind regards,

Visalini (Lini) Nair‑Shalliker

Academic Editor

PLOS ONE

Reviewer's Responses to Questions

**Comments to the Author**

Reviewer #1: All comments have been addressed

2. Is the manuscript technically sound, and do the data support the conclusions?

Reviewer #1: Yes

3. Has the statistical analysis been performed appropriately and rigorously?

Reviewer #1: Yes

4. Have the authors made all data underlying the findings in their manuscript fully available?

Reviewer #1: No

5. Is the manuscript presented in an intelligible fashion and written in standard English?

Reviewer #1: Yes

Reviewer #1: Thank you for carefully addressing my previous comments and suggestions. I found the revised manuscript to be easier to follow and the additional content improved my understanding of the research study. I have two additional comments based on reading the revised manuscript:

1. The last paragraph on page 13 of track-changed version of the manuscript seems to require clarification in order to understand what is being presented. This is the sentence that seems to need some additional explanation:

“The majority of these participants (61%, n=81) were not offered a choice of collection method, however 38% (n=50)

participants were offered a choice and elected to have a clinician-collected sample, and an additional

two couldn’t remember.”

2. On page 20, this sentence would benefit from the addition of the word 'group': 21 Interestingly, the majority of this GROUP had previously screened and ...”

**Do you want your identity to be public for this peer review?** For information about this choice, including consent withdrawal, please see our Privacy Policy

Reviewer #1: No

---

## [Author Response · Author response to Decision Letter 2]

7 Feb 2026

Thank you for your thoughtful feedback and consideration of our manuscript. We submit a revised version incorporating the recommendations. Our response is provided in the document "Response to Reviewers".

---

## [Editor Report · Decision Letter 2]

23 Feb 2026

Experiences of HPV self-collection among Aboriginal and Torres Strait Islander women and people with a cervix

PONE-D-25-29330R2

Dear Dr. Mitchell,

We’re pleased to inform you that your manuscript has been judged scientifically suitable for publication and will be formally accepted for publication once it meets all outstanding technical requirements.

Kind regards,

Visalini Nair-Shalliker

Academic Editor

PLOS One
---

## [Editor Report · Acceptance letter]

PONE-D-25-29330R2

PLOS One

Dear Dr. Mitchell,

I'm pleased to inform you that your manuscript has been deemed suitable for publication in PLOS One. Congratulations! Your manuscript is now being handed over to our production team.

Kind regards,

on behalf of

Dr. Visalini Nair-Shalliker

Academic Editor

PLOS One